# Deriving Lipid Classification Based on Molecular Formulas

**DOI:** 10.3390/metabo10030122

**Published:** 2020-03-24

**Authors:** Joshua M. Mitchell, Robert M. Flight, Hunter N.B. Moseley

**Affiliations:** 1Department of Molecular & Cellular Biochemistry, University of Kentucky, Lexington, KY 40536, USA; jmmi243@uky.edu (J.M.M.); robert.flight@uky.edu (R.M.F.); 2Markey Cancer Center, University of Kentucky, Lexington, KY 40536, USA; 3Resource Center for Stable Isotope Resolved Metabolomics, University of Kentucky, Lexington, KY 40536, USA; 4Institute for Biomedical Informatics, University of Kentucky, Lexington, KY 40536, USA; 5Center for Clinical and Translational Science, University of Kentucky, Lexington, KY 40536, USA

**Keywords:** SMIRFE, lipidomics, metabolomics, lipid category, machine learning, Random Forest

## Abstract

Despite instrument and algorithmic improvements, the untargeted and accurate assignment of metabolites remains an unsolved problem in metabolomics. New assignment methods such as our SMIRFE algorithm can assign elemental molecular formulas to observed spectral features in a highly untargeted manner without orthogonal information from tandem MS or chromatography. However, for many lipidomics applications, it is necessary to know at least the lipid category or class that is associated with a detected spectral feature to derive a biochemical interpretation. Our goal is to develop a method for robustly classifying elemental molecular formula assignments into lipid categories for an application to SMIRFE-generated assignments. Using a Random Forest machine learning approach, we developed a method that can predict lipid category and class from SMIRFE non-adducted molecular formula assignments. Our methods achieve high average predictive accuracy (>90%) and precision (>83%) across all eight of the lipid categories in the LIPIDMAPS database. Classification performance was evaluated using sets of theoretical, data-derived, and artifactual molecular formulas. Our methods enable the lipid classification of non-adducted molecular formula assignments generated by SMIRFE without orthogonal information, facilitating the biochemical interpretation of untargeted lipidomics experiments. This lipid classification appears insufficient for validating single-spectrum assignments, but could be useful in cross-spectrum assignment validation.

## 1. Introduction

Lipidomics is the subdiscipline of metabolomics concerned with the analytical investigation of the lipidome, the set of lipid metabolites within a (biological) sample, and their roles within the metabolome. Unlike other categories of metabolites that are largely grouped based on their structures, lipids are defined by their very low solubility in water and collectively represent a structurally and chemically diverse set of metabolites with various roles in normal and pathological cellular function. Because of this structural and chemical diversity, which often confers amphipathic properties, seemingly every life process involves lipids, including but not limited to the maintenance of cellular structure [1]; membrane fluidity [2,3]; intracellular, extracellular, and hormonal signaling [4,5]; energy metabolism [6,7]; disease processes [8], including cancer [9,10]; and aging [11,12]. Thus, through lipidomics, more complete modeling of cellular metabolism and a better understanding of physiological and pathological processes at the mechanistic level can be achieved [13].

Although the potential benefits of lipidomics are enormous, the rigorous analytical investigation of the lipidome in real-world biological samples requires the reliable observation of lipid features in the samples as well as the accurate assignment of those features to a lipid structure and/or lipid class. This represents a significant bioanalytical chemistry problem due to the high structural diversity of lipids, their wide range of observed concentrations, and differences in lipid profiles across compartments and time [14,15,16]. Due to its sensitivity to a wide range of chemical structures and low detection limits, mass spectrometry remains the most popular analytical technique for lipidomics analysis [17]. Traditionally, mass spectrometry has been used in conjunction with other analytical techniques, such as gas chromatography [18], liquid chromatography [19,20], or TLC [21], to provide additional orthogonal information that aid in the assignment of observed lipid features. Recent advances in mass spectrometry, namely Fourier transform mass spectrometry (FT-MS), have provided significant improvements in mass accuracy, mass resolution, and sensitivity [22]. Together, these analytical improvements provide the capability to resolve distinct isotopologues with identical unit masses, which in turn enables the analysis of multi-isotope labeling experiments [23,24], improved assignment accuracy without orthogonal chemical information [25], and the detection of compounds in the sub-femtomolar range [26]. These capabilities enable the use of stable isotope resolved metabolomics (SIRM) techniques in combination with traditional lipidomics methodologies [27] that can provide richer information, allowing the elucidation of unknown metabolic pathways, the quantification of relative fluxes through connected metabolic pathways, and the identification of active metabolic pathways under various cellular conditions [28,29]. Notably, improvements in FT-MS have enabled significant advances in shotgun lipidomics by distinguishing and quantifying isobaric lipids in place of in-depth MS/MS [30]. When combined with MS/MS, the analytical advantages of FT-MS enable global lipidome analysis and simultaneous structural characterization of some detected lipids [31]. However, the full utilization of the analytical capabilities of FT-MS for lipidomics, especially untargeted lipidomics, will require the development of new data analysis methods better tailored to the data provided by FT-MS.

Although mass spectrometry, especially FT-MS, enables the robust *detection* of lipid features, the assignment of those features to lipids and by extension to lipid class, remains challenging. Moreover, mass spectrometry can also detect previously unobserved lipids; however, existing spectral assignment methodologies, such as LipidSearch [32] and PREMISE [33], rely heavily or exclusively on observed *m/z* values from MS1 to query databases of known metabolites for assignment. This can result in either a lack of assignments for these features or worse, incorrect assignments for these features, which can cause large interpretive errors later in analysis. The presence of spectral artifacts in FT-MS spectra can result in the consistent misassignment of artifactual features, leading to substantial errors in downstream analyses [34]. Additionally, the potential for assignment bias from using these databases significantly hampers the potential for discovery [35]—a stated goal of many untargeted lipidomics analyses. Orthogonal information from chromatography and/or MS/MS can cross-validate potential lipid assignments [36]; however, the necessary combined analytical setups are more complex, generate additional information that must be processed, require larger amounts of sample [37], are incompatible with direct infusion experiments, and can still suffer from assignment bias when using fragmentation or retention time annotated databases. Furthermore, tandem MS is not as sensitive as MS1. Moreover, the incompleteness of metabolite and lipid databases [38,39] is a major source of assignment error that cannot be easily overcome through additional orthogonal information.

Since orthogonal chromatographic information is not a panacea for metabolite detection and assignment, advances in FT-MS assignment techniques and improvements in electrospray ionization have made direct-infusion mass spectrometry an increasingly popular analytical setup for both metabolomics and lipidomics. Using assignment techniques, such as our in-house SMIRFE algorithm [25,40], elemental molecular formulas can be robustly assigned to observed spectral features without information from chromatography or MS/MS and without querying existing databases of metabolite and lipid structures. Our methodology searches a near-exhaustive CHONPS elemental formula search space to find possible molecular formulas represented in a spectrum and then filters to a set of likely assignments by comparing observed intensity ratios between isotopologues of those molecular formulas [25]. This assignment methodology is ideally suited for untargeted metabolomics and lipidomics workflows, with the molecular formula assignments useful for both biomarker characterization and relative metabolic flux analysis after natural abundance correction [23,41]. Furthermore, this assignment methodology is resilient against misassignment due to common artifacts in FT-MS spectra. However, many lipidomics experiments are concerned with changes at the lipid class level, which neither SMIRFE molecular formula assignments [25], elemental analysis by other methods like inductively coupled plasma mass spectrometry [42,43], nor van Krevelen diagram approaches [44,45] can directly provide.

Lipid classes are sets of lipids that share certain chemical structure features. When the chemical structure of a lipid feature is known, either through database lookup or other analytical approaches, the classification of that structure into a lipid class is straightforward. Automated tools, such as ClassyFire [46], use machine learning methods to automatically assign lipid class (and more generally metabolite class) based on the input structures; however, structural information cannot be directly acquired through elemental molecular formula assignment methods. Although the detection of potential lipid features can be achieved using ratios of heteroatoms [47], the classification of non-adducted molecular formulas into specific lipid classes remains an unsolved problem that can prevent the effective biochemical interpretation of SMIRFE-generated non-adducted formulas derived from class-level lipidomics analyses. In this manuscript, we present a novel method of predicting lipid category and class from molecular formula and information readily available from direct-infusion MS1 spectra without reliance on existing metabolite databases or orthogonal information.

Manually constructing rules that can map elemental molecular formulas to lipid classes without additional orthogonal information is a daunting proposition and would result in rules that are fragile, likely incomplete, and incorrect. Fortunately, the prediction of lipid class from compound properties derivable from MS1 spectra, namely their elemental molecular formula, can be stated as a supervised machine learning problem. In supervised machine learning, models are trained that predict a ‘label’ (e.g., lipid class) from a set of features (i.e., a feature vector) describing an input (e.g., elemental components of a molecular formula). These models are not constructed by hand, but rather, example inputs with known labels are used to ‘train’ a model.

Using a large lipid database such as LIPIDMAPS [48] that contains many examples of known lipids and their associated lipid classification, a set of generalized predictive models can be constructed (i.e., trained) to infer rules for predicting the correct lipid classes from elemental molecular formulas. The LIPIDMAPS database is the largest lipid-specific repository of metabolite structures and every entry in LIPIDMAPS is assigned to both a high-level lipid category and a lower-level lipid class. There are eight lipid categories, which are further subdivided into 79 distinct classes. Each entry represents either an observed lipid or a predicted lipid and contains an elemental molecular formula for that lipid and its assigned lipid category and lipid class. Therefore, entries from LIPIDMAPS are sources of true positives for our training dataset. LIPIDMAPS is also subdivided into two databases: the LIPIDMAPS Structure Database (LMSD) and the LIPIDMAPS In-Silico Structure Database (LMISSD). True negatives are also needed for the construction of robust models. In this case, a true negative is a biological formula that is not a lipid. The human metabolome database (HMDB) [49] contains many examples of biological formulas of known class and is freely downloadable. By removing known lipids from the HMDB, a set of false negatives can be constructed.

As shown previously [38], isomerism is common with all metabolites but especially with lipids. This is reflected in both the LMSD and LMISSD, which contain only 7.4% and 0.053% non-isomeric entries. See Appendix A for the level of isomerism broken down by category and class in LMSD and LMISSD, respectively. In the case of the LMISSD, this high amount of isomerism likely reflects both the high isomerism of lipids as a metabolite class as well as the methods used to generate the additional entries for the database in silico. Moreover, de-duplicating isomeric formulas is a necessary step in constructing a training dataset in order to prevent deleterious training effects from duplicate molecular formula entries.

The selection of both a chemically-descriptive feature vector and an appropriate machine learning algorithm will heavily influence the performance and applicability of the resulting lipid class predictive models. Feature vectors must be sufficiently descriptive so that the algorithm has enough information to differentiate between inputs with different lipid class labels, but must also be limited to information that can be readily and accurately acquired through direct infusion FT-MS MS1 experiments. As such, structural information, which is the most informative, cannot be included in our feature vectors. Limiting our feature vectors to only information that can be acquired routinely from MS1, namely, non-adducted elemental molecular formula assignments provided by our SMIRFE algorithm, still provides substantial chemical information, including the number of atoms for each element present in the formula and the theoretical monoisotopic mass.

Random Forest [50] has been successfully applied to many metabolomics problems [51,52] and has several properties that make it an ideal machine learning method for this use case. First, the classification of inputs into lipid classes is a highly hierarchical problem for which Random Forest provides excellent performance. Second, a Random Forest of decision trees excels at learning classification rules based on discrete data like elemental atom counts. Third, the bagging process intrinsic to the Random Forest algorithm protects against overfitting and enables the direct measurement of classifier accuracy similar to explicit cross-validation [53]. Fourth, bagging and the construction of many independent binary classifiers makes unbalanced training datasets, where each label is not equally represented in the training data, less problematic. This last property is especially important since a training dataset based on LIPIDMAPS will be highly unbalanced with respect to the different lipid classes (see Table 1 and Table 2).

## 2. Results

### 2.1. Monolithic Classifier Performance on Training Datasets

Using the LMSD + HMDB_non_lipid dataset, the performance of a monolithic classifier for lipid category and lipid class was evaluated. Even with 500 trees, the monolithic Random Forest models were only able to achieve an out-of-bag accuracy of 74.9% for lipid category and 87.3% for lipid class. Including the LMISSD resulted in an out-of-bag accuracy of 83.1% for lipid categories and 80.1% for lipid class (confusion matrices for each classifier can be found in Appendix A). In both datasets, the presence of a large number of non-lipid entries inflates the lipid class accuracy as all non-lipid entries map to the non-lipid class. Although monolithic classifiers may have the theoretical advantage of being simpler to implement, train, and deploy, their usefulness is limited by their relatively poor classification performance.

### 2.2. Multi-Classifier Performance on Training Datasets

For both training datasets (LMSD + HMDB_non_lipid and LMSD + LMISSD + HMDB_non_lipid), the out-of-bag accuracy and precision for each lipid category are shown in Table 1, while the class level results are shown in Appendix A. For all categories, the LMSD + HMDB_non_lipid trained models achieved high precision and high accuracy for all lipid categories. Classification performance for lipid class varies between classes but is in general excellent for classes with enough entries. The LMISSD-trained models achieved similar precision and accuracy for all categories (Table 2) and classes (Appendix A). Although individually high accuracy or high precision would not necessarily indicate a well-trained model, the combination of high accuracy and precision across the models implies that the combined classification performance is robust and can be effectively applied to experimentally-derived molecular formulas.

The category-level models trained on the LMSD + HMDB_non_lipid dataset demonstrate excellent accuracy on all categories and excellent precision for all categories apart from polyketides (76.7%). The polyketides represent a very diverse set of structures compared to other lipid classes which explains this discrepancy. The number of entries of each category highlights the unbalanced nature of this dataset and motivated the use of Random Forest for these models. Each model was trained as a one-class against all others model (i.e., the Fatty Acyl [FA] model was trained using the set of known Fatty Acyls as true positives and all other entries as true negatives). Inclusion of the LMISSD provided no additional examples of Fatty Acyls, Polyketides, Saccharolipids, or Sterol Lipids and had minimal effect on the precision and accuracy of the models.

### 2.3. Multi-Classifier Performance on Theoretical Molecular Formulas

Brute force enumeration and testing of all points within the convex hull constructed around all CHONPS-only molecular formulas in the HMDB identified 110,857,519 formulas. While a brute force approach was computationally expensive, requiring several thousand CPU-core hours, it was necessary due to memory constraints with more complex methods. Classification of the convex hull formulas took approximately 10 CPU-core hours. Calculations were performed on a quad-socket system with four E7-4820v4 CPUs (10 cores, 20 threads each) clocked at 2.00 Ghz and 1TB of RAM clocked at 2400 MHz. Classifying these formulas with the LMSD and LMISSD models resulted in the majority of formulas assigned to either the non_lipid category or to no category at all. Results for each category are summarized in Table 3 and Table 4 and by class in Table 5 and Table 6. The LMISSD models predict four of the seven categories more frequently than the LMSD models but the trend in predicted categories was similar. Due to the number of formulas in the convex hull that do not correspond to ‘real’ metabolite formulas, a high percentage of non_lipid or no classification formulas is expected if our models are highly discriminating. Similar results were observed at the class-level (Appendix A).

The formulas within the convex hull surrounded by the HMDB metabolites represent a very large set of plausible metabolites formulas. Lipid categories were predicted for every formula within the hull. For all categories, more formulas were predicted for each category than existed in the training dataset, indicating that the models have generalized beyond the training dataset. The extent of this generalization varied depending on the training dataset. For example, saccharolipids (the category with the smallest number of entries in the training dataset) was predicted more frequently in the LMSD-trained models than in the LMISSD models, while sphingolipids were more frequently predicted in the LMISSD-trained models than in the LMSD-trained models. Although the distribution of predicted lipid categories varies slightly between the two sets of models, the overall trends are comparable. For example, sphingolipids were the highest predicted lipid category from the convex hull dataset by both sets of models.

### 2.4. Multi-Classifier Performance on Experimentally-Observed Molecular Formulas

The distribution of the assigned lipid categories on molecular formulas enumerated from a human lung cancer FT-MS dataset is shown in Table 5 for the LMSD-based classifier. SMIRFE generates many possible assignments for each peak at higher *m/z* as the number of possible formulas increases dramatically with increasing *m/z*. As a result, a relatively small percentage of formulas are assigned to a lipid category but many peaks have at least one formula that was assigned to a lipid category. For the LMSD models, the ability to predict lipid category and class drops substantially after about 1200 *m/z*. This is due to the low number of entries in the LMSD at higher *m/z*.

When the masses of the peaks are shifted by +21 *m/z* to mimic a gross miscalibration error, the number of SMIRFE assignments is increased from 127,338 to 131,690 total formulas and the number of predicted lipids increases as well from 32,688 to 34,755 (Table 5 and Table 6). This result implies that the lipid classifier alone cannot be used to screen out all bad assignments when lipids are expected. Instead, other orthogonal data must be used to verify the quality of the assignments and select the correct assignments.

SMIRFE assignments were generated for the non-small cell lung cancer (NSCLC) dataset described in the Appendix A. SMIRFE assignments are generated in an untargeted manner without using a database of known lipids. For the peak masses across all peaklists, 127,338 total formulas were assigned and then classified (this total includes everything and does not represent any filtering based on E-value). A total of 32,688 lipid category classifications were made with the most commonly assigned categories being glycerophospholipids and sphingolipids. A similar result was observed in the convex hull results as well, potentially indicating that this is an artifact of the classification method or possibly that these lipid categories are much more diverse than other categories. When each peak was shifted by 21 *m/z*, roughly 3% more formulas were assigned and 6% more lipids were classified. This small relative increase in SMIRFE assignments is likely due to the increased search space density with a 21 *m/z* shift. More importantly, the large number of artifactual assignments reflects the necessity of high-quality data before classification. Methods that can predict high-quality assignments correctly are not necessarily protected from the effects of low-quality spectral data that can cause misassignment.

### 2.5. Cross-Sample Assignment Correspondence along with Lipid Classification Improves Assignment Quality

Limits in mass and intensity resolution coupled with the immense size of the search space considered by SMIRFE at high masses leads to ambiguous assignments for many peaks. When mass error is present, ambiguous and incorrect assignments can be generated. However, the correct assignment for a peak should be assigned more consistently for a consistently observed feature in the dataset. Therefore, how well an assignment corresponds across samples in a dataset is a potential avenue for selecting high-quality assignments. Figure 1 shows histograms of assignment correspondence at an E-value <= 0.1 for lipid-classified elemental molecular formulas (EMFs) derived from the spectra of the lung cancer dataset. Much higher correspondence is observed in the unshifted assignments vs the shifted assignments and the number of shifted high-correspondence assignments are much fewer at lower *m/z*. This trend is even stronger with an E-value <= 0.01 as seen in Appendix A. Without lipid classification or a low E-value cutoff, the trend is not as strong at higher *m/z* (see Appendix A). These results imply that both lipid classification and assignment correspondence can be used to filter out incorrect assignments.

## 3. Discussion

### 3.1. Classifier Organization and Performance

The hierarchical model organization has several distinct advantages over the monolithic implementation. First, the hierarchal organization enables the simplification of each decision boundary that each model must learn and each model can select its optimal set of features for determining that boundary. Second, using category classifiers to filter what feature vectors should be passed to lower-level class models effectively results in machine learning models feeding their results into other machine learning models. This technique is employed in deep learning to construct robust and powerful classifiers for complicated classification problems. Third and finally, collections of relatively weak classifiers working together often outperform monolithic classifiers. This observation is also well-known in the machine learning field [54] and is the central motivating concept behind ensemble machine learning algorithms, such as Random Forest. These advantages come at the cost of additional manual overhead to segment and organize the training datasets appropriately and additional computational overhead to construct and train multiple models. This cost is largely mitigated by the fact that models need to be trained only once (or very rarely) and can then be reused, effectively amortizing the cost of building the models and using preconstructed models for classification is computationally cheap.

As mentioned previously, the monolithic, multi-lipid-class predictive model failed to achieve top performance for the task of classifying assigned molecular formulas into lipid categories and classes. We hypothesize that this is due to the inability of a single classifier to represent all decision boundaries completely and accurately. This single classifier must not only learn how to separate lipids from non-lipids, but it must also subdivide the lipid feature space into discrete spaces representing each category and further subdivide these category spaces into class spaces. Much of this subdivision can be done explicitly during training. For example, the diacylglycerols are a sub-class of the larger category of glycerolipids and a less powerful classifier can easily identify the diacylglycerols from other glycerolipids when it must only learn that single decision boundary. As a result, our organization of weaker predictive models had superior performance. Initially, this behavior can seem counterintuitive but is consistent with the concept of ensemble learning from machine learning where collections of weaker classifier models often outperform fewer larger classifier models when properly organized. The hierarchy of models that are constructed mirror how a human would approach the classification problem. For example, if a molecular formula is known not to be of the sphingolipid category, a human will not attempt to assign this formula to a sphingolipid class; however, a monolithic model will attempt to do so. This wastes computational power and increases the likelihood of incorrect prediction of both class and category.

The final models produced by our tool achieved both high accuracy and high sensitivity on the training dataset. Of course, performance on training data does not paint a complete picture of model performance, but for Random Forest which implements bagging, these metrics predict performance on inputs similar to the training data. Models with both high accuracy and high sensitivity are unlikely to produce incorrect lipid assignments, but may be overly conservative and fail to generate a non_lipid assignment for some inputs. While this behavior is undesirable, it is preferable to less conservative models that will yield many incorrect lipid category and class predictions.

### 3.2. LMSD vs LMISSD Trained Models

One method for improving the performance of a machine learning model is to provide larger amounts of training data, which in turn enables more informed and more accurate decision boundaries to be determined. For this reason, models were trained using both the LMSD and LMISSD, which has nearly 25 times the number of entries as the LMSD. However, LMISSD-trained models did not offer substantially improved performance as compared to the LMSD-only models on the training datasets. Although the LMISSD contained many entries, the input training set only doubled in size after isomeric entries were removed, implying that little information was added regarding the distribution of formulas in lipid category or lipid class space. Another possible explanation for this observation is that the LMISSD contains substantially more entries, but for only four out of the seven categories in the LIPIDMAPS database and that the decision boundaries for these categories were already well-determined by the LMSD-only models.

### 3.3. Classifier Generalization

A benefit that machine learning models have over traditional database lookups are their ability to infer rules that can be applied to never observed inputs to make accurate predictions. However, testing this ability to generalize is not easy without a gold standard dataset. The closest gold standard dataset is LIPIDMAPS itself; however, this was used for classifier training. Other lipid databases like LipidHome [55] do not contain enough entries for all 8 lipid categories and 79 lipid classes for rigorous testing of generalization. Therefore, we constructed a convex hull of theoretical biologically-relevant molecular formulas around the HMDB to evaluate both the ability to generalize, but also the potential for over-generalization that would lead to large false-positive classification. The ability to generalize was demonstrated with both the LMSD- and LMISSD-trained models. Both models produced lipid category and class predictions for experimental and theoretical molecular formulas not present in the training dataset. However, the generalizability of the models depends heavily on the quality and size of the input dataset (Appendix A).

As shown in Table 5 and Table 6 and Appendix A, the LMISSD- and LMSD-trained models had similar behavior on the convex hull metabolites. In general for both sets of trained models, the categories and classes with more entries in the training dataset were more frequently predicted. Overall, the percentages of hull formulas predicted for each category and class were similar between the two models, implying that the two sets of models are very similar. Both sets of models predicted roughly the same number of non_lipid formulas implying that the overall lipid vs non-lipid decision boundaries of the two models are very similar. However, a high percentage of sphingolipids were predicted by both sets of models. This probably represents an over-generalization for sphingolipid classification due to a bias in the trained models from the unbalanced training data, but it could reflect the relative amount of structural diversity possible within each category, i.e., the number of possible sphingolipid formulas, in particular acidic glycosphingolipids, might truly be much larger than the number of possible sterol formulas. Moreover, the LMISSD models assigned almost twice as many sphingolipids as the LMSD models (11.34% vs 6.761% of the convex hull). Likewise, at the class level (see Appendix A), the LMISSD models assigned almost twice as many acidic glycosphingolipids [SP06] as the LMSD models (5.096% vs 2.878%). However, the performance of both sets of models are very similar for the other sphingolipid classes and are a relatively low percent of the convex hull except for the neutral and acidic glycosphingolipids. Discrepancies between the two models can also be attributed to the presence of predicted lipids in the LMISSD that do not exist—this could confuse classifiers if the predicted lipids and the validated lipids suggest different decision boundaries. However, both the LMISSD and LMSD training sets have the same number of acidic glycosphingolipids examples (i.e., 520 examples). Therefore, the over-generalization of this class is likely coming from the addition of other lipid examples in the LMISSD training set.

Ultimately, the ability of both models to make accurate predictions will be improved with larger training datasets. With more entries that more exhaustively span the lipid formula space, the more accurate and generalizable the models constructed using these same methods will become. However, due to the marginal improvement with a doubling of the training dataset from the LMSD vs LMISSD, improvements may still be marginal without a vastly larger training dataset.

### 3.4. Mass Error and Classification Results

Ideally, a substantial mass error would result in no formulas being assigned by SMIRFE or that the assigned formulas fail to classify. As shown with our NSCLC dataset, a large mass error does not eliminate all assignments nor completely abolish our ability to classify the resulting, almost certainly incorrect, assigned formulas.

Due to the very large search space that an untargeted tool must search to generate assignments, almost any *m/z* has many possible assignments, due to the theoretical molecular formula search space. Since a systematic error does not change the mass difference between isotopologues, patterns of isotopologues for these incorrect formulas can still be identified and assigned. Thus, without extremely high mass resolution to restrict the set of possible assignments considerably, which still may not be effective [56], a constant mass error will still produce assignments. Furthermore, the current variance in peak intensities achievable with current-generation instruments is not low enough to prevent artifactual assignment at higher *m/z*.

As was seen in the convex hull analysis, approximately a quarter of the generated formulas appeared to be lipids to the models. This could reflect the true distribution of lipids in possible formula space, but more represents the limitations of our models. Nonsense formulas that can arise from *m/z* error or from the convex hull method cannot be properly learned as they are very different from the training set data. Although the ability of our models to produce no classification for an input feature vector protects against this effect, it is not perfect. The same models that learn real (biochemically relevant) metabolite formulas correctly may fail to properly handle nonsense formulas that SMIRFE can assign to peaks with high mass error and noise or artifactual peaks. Therefore, lipid classification alone should not be used to filter out features in datasets, especially on a single-spectrum basis. Information such as how many times a formula is observed across a dataset appears to be very useful for filtering. In particular, combining E-value, assignment correspondence, and lipid classification appears very useful for improving assignment quality. Moreover, the observed correlation between features classified to the same lipid category and/or class should provide additional discriminating criteria. Features considered trustworthy by this information and other methods can then be used for further analysis. Similar problems exist with targeted assignment tools as well and a lack of substantial cross-sample formula correspondence is an indication that there is a possible data quality problem preventing accurate assignment.

### 3.5. Implications for Experimental Design

The ability to predict lipid category and class from molecular formula assignments without the need for cross-validated *metabolite* assignments enables simpler experimental designs as the volume of information needed to perform class or category level comparisons is lessened. As molecular formulas can be assigned from direct infusion FT-MS MS1 spectra directly and in a cross-validating manner, chromatography and other orthogonal information are not necessary for class or category level comparisons when using these models. However, the quality of the analyses will depend heavily on the quality of the assigned molecular formulas.

SMIRFE leverages patterns in the relative heights of isotopologue peaks for the same elemental molecular formula to determine what molecular formulas best explain features observed in a spectrum. Although SMIRE is not necessarily limited to only high-end mass spectrometers, such as FT-MS instruments, only these instruments provide enough mass accuracy and resolution to observe and characterize relevant sets of isotopologues. This restriction is becoming increasingly less relevant as high-performance spectrometers become more available. Additionally, SMIRFE and subsequent lipid prediction do not enable the robust assignment of metabolite structures to spectral features and this will still require additional information from orthogonal experiments.

## 4. Materials and Methods

### 4.1. Structure of Chemically-Descriptive Feature Vectors

As illustrated in Figure 2, the feature vector, based on a given molecular formula, contains an atom count for each CHONPS element, the sum of atom counts for other elements, as well as the theoretical monoisotopic mass and individual decimal places from this mass. These features were selected for final classifier training based on average Random Forest Gini feature importance scores from several earlier rounds of classifier building and testing that utilized a larger set of features derived from the given molecular formula. To ensure that all molecular weights for all entries were consistently calculated and thus maximally comparable for classification, every entry had its theoretical monoisotopic molecular mass re-calculated using isotope molecular masses from NIST [57,58]. Each element atom count is an integer, but for different elements the expected atom count range can vary significantly. For biological lipids in general, up to 300 hydrogen atoms could be expected, but only a few sulfur or phosphorous atoms are expected. The theoretical monoisotopic mass is a floating-point number between zero and a few thousand Daltons, while each digit will be represented as an integer between 0 and 9. As a result, each feature in our feature vector will be on a different scale. Although these could be normalized to remedy the differences in scale, which is a requirement for some machine learning algorithms, the Random Forest algorithm does not have this limitation.

### 4.2. Derivation and Organization of Training Datasets

In addition to the selection of proper feature vectors and the selection of an appropriate machine learning algorithm, the quality of a machine learning model depends heavily on the quality of the training data from which the model is constructed. Training datasets should be large, contain examples of both true positives and true negatives, and cover most of the expected feature space. Additionally, training data must be organized appropriately. In this case, the training data should have the training inputs mapped to both high-level lipid categories (e.g., glycerolipid, phospholipid, etc.) and further subdivided into more specific “main classes” (e.g., monoradylglycerols, eicosanoids, secosteroids, etc.).

The LMSD contains both manually verified and computationally generated lipids and is freely available for download, while the LMISSD is composed of completely computationally generated lipids. Unlike the LMSD, the LMISSD is not directly downloadable and a web scraper written in R [59] using the RSelenium package [60] was used to extract every LMISSD entry with its lipid category, lipid class, and molecular formula. We downloaded the LMSD in September 2018, which contained 42,004 entries. We scraped the LMISSD in September 2018, obtaining 1,131,106 entries.

We also downloaded version 4.0 of the HMDB in September 2018, which contained 114,089 entries with 22,657 entries being non-lipids. By filtering out and removing known lipids from the HMDB, a set of false negatives was constructed. These entries, of course, do not have a lipid category or lipid class assigned to them, thus an extra category and class called ‘non_lipid’ was assigned to these entries.

Since in-silico generated lipids may not necessarily exist in biological systems, it is prudent to construct two example training datasets: HMDB non_lipids + LMSD (referred to as LMSD training set) and HMDB non_lipids + LMSD + LMISSD (referred to as LMISSD training set). Since isomers of lipids can have the same molecular formula but have a different structure that can even belong to different lipid categories and lipid classes, each training dataset was de-duplicated by mapping each formula to all observed lipid categories and classes for each formula. A large portion of the entries in both the LMSD and LMISSD are isomers of other entries of the same lipid class and category (see Appendix A). The final LMSD + HMDB non_lipid training dataset resulted in 16,215 unique entries as compared to 30,692 for the LMSD + LMISSD + HMDB non_lipid training dataset.

### 4.3. HMDB-Derived Molecular Formula Convex Hull Construction

From the set of HMDB formulas composed only of CHONPS elements and with a molecular weight below 1600 *m/z*, a convex hull was constructed and enumerated to generate theoretical metabolite formulas of biological origin. In this formulation, each molecular formula from the HMDB represents a point in a six-dimensional space (each dimension representing the number of a CHONPS element present in the formula) where each point has integer coordinates corresponding to the number of each element present. A convex hull around these points was constructed using the Python implementation (version 2015.2.1) of the qhull algorithm downloaded from the Python Package Index [61]. All possible points within the convex hull were then enumerated to generate all CHONPS-specific molecular formulas within the hull.

### 4.4. Experimentally-Derived Molecular Formulas from Human Lung Cancer Samples

Paired cancer and non-cancer tissue samples were acquired from eighty-six patients with suspected resectable stage I or IIa primary non-small cell lung cancer (NSCLC). Specimens were obtained primarily using wedge resection and all specimens were harvested within 5 minutes after pulmonary vein clamping to minimize ischemia in the resected tissues. Immediately after resection, the tumor was transected and sections of cancerous tissue and surrounding non-cancer tissue at least 5 cm away from the tumor were immediately flash-frozen in liquid nitrogen and stored at −80 °C. On-site pathologists confirmed the diagnosis and cancer-free margins on parallel tissue samples. All samples were collected under a University of Louisville approved Internal Review Board protocol and written informed consent was obtained from all subjects before inclusion in the study. The frozen samples were then prepared and analyzed using two Thermo Orbitrap Fusion instruments interfaced to an Advion Nanomate nanoelectrospray source. Additional details on sample preparation and mass spectrometric analysis are included in Appendix B.

MS1 spectra were acquired for each sample using direct infusion. These MS1 spectra were then assigned using our in-house SMIRFE assignment tool which assigns spectral features without a database of expected molecular formulas corresponding to metabolites. Instead, SMIRFE generates an exhaustive list of expected molecular formulas which can be queried using a peak’s observed *m/z* with a mass tolerance determined by the digital resolution of the instrument, which is approximately 1 ppm for the Fusion instrument. SMIRFE uses patterns in the intensity ratios of suspected isotopologues of the same EMF and how these patterns compare to predicted intensity ratios based on isotope natural abundances. To account for sources of intensity error, such as ion suppression and scan-to-scan injection efficiency differences, when SMIRFE compares the intensity of two peaks in a spectrum, it calculates a median intensity ratio across all the scans containing both peaks. Additionally, the log variance on that ratio across scans is calculated and used to inform the comparison of the observed intensity ratio to the predicted [25]. Other strategies, such as scan normalization, the removal of artifactual peaks, and the removal of outlier scans also help mitigate other sources of error.

Assignments were generated for 192 samples up to 1600 *m/z* and SMIRFE assigned a total of 127,338 unique formulas. This is a large number of assignments; however, this includes multiple possible assignments for each peak and does not account for any filtering of assignments based on their E-values.

### 4.5. Classifier Construction

Initially, a single, monolithic Random Forest classifier was constructed for the simultaneous classification of all lipid categories and classes. This construction was done using the Random Forest implementation from sklearn [62] with default hyperparameters except for the number of decision trees, which were varied from the default of 10 trees to 500 trees.

Using the monolithic organization, a single model exists for lipid categories and a second model exists for lipid classes (Figure 3A). Each query feature vector is processed by both models to produce category and class labels and each model can be used independently. While conceptually simple and easier to implement, the monolithic organization suffers from relatively poor performance.

An alternative approach is a hierarchy of models in which category and class models are combined hierarchically (Figure 3B). Each class and category has a separate Random Forest model, but the category-level classification dictates the use of class-level classification. For all models, the Random Forest implementation from the Python sklearn package v0.22 [62] was used with default hyperparameters except for the number of trees which was set to 500.

### 4.6. Evaluation of Lipid Classification Performance

The performance of any machine learning model can be evaluated using a variety of metrics; however, rarely does a single metric fully reflect the goodness of any model in all use cases. For example, classifiers with high overall accuracy for the whole training set may classify certain labels very poorly which may not be obvious from a global accuracy metric. This is especially true for unbalanced training datasets, where conservative models will have high accuracy but make very few classifications. Thus, models that are both accurate and highly specific for all lipid categories and classes individually are highly desirable. Models that generate many incorrect assignments will, at the very least, become burdensome to be utilized effectively, and at worst, could lead to incorrect interpretation of results. Therefore, highly accurate and highly precise models are desirable even at the cost of missing some true positive classifications.

To evaluate the accuracy of the Random Forest models, we used the out-of-bag training accuracy. While not strictly equivalent to explicit cross-validation, the accuracy metric provided by out-of-bag training accuracy is a sufficient proxy. In fact, for unbalanced datasets such as ours, the out-of-bag training accuracy can *underestimate* the error rate [63]. Therefore, the use of k-fold cross-validation with Random Forest would only reduce training sample size and could substantially overestimate the error. For precision, the classic definition of precision can be applied for sets of inputs of known label and their model-generated labeling. Models that are both highly accurate and highly precise are unlikely to generate false assignments and are suitable for our task.

## 5. Conclusions

With untargeted analysis methods, lipidomics has the potential to produce more informative datasets that will aid in the construction of more complete models of cellular metabolism. This in turn enables a better understanding of both healthy and disease processes. A necessary step in many of these analyses is the assignment of lipid category or class to an observed lipid feature. When multiple orthogonal sources of information are available (i.e., MS + chromatography, NMR + chromatography, MS/MS), lipid category and class assignment can be inferred from trustworthy metabolite assignments based on comparison to spectral databases; however, this approach limits untargeted analysis, since spectral and lipid databases are incomplete and NMR and MS/MS detect features for far fewer metabolites than MS1.

The application of machine learning algorithms enables the construction of models that can accurately and precisely assign lipid labels to observed spectral features that have been assigned to a molecular formula. Unlike other approaches that leverage metabolite databases directly for lipid assignment, these models can infer lipid category and class for entries not present in existing databases. This capacity is essential for untargeted metabolomics experiments as database incompleteness can lead to a biasing of lipid classification and in turn biological interpretation. Since these models are informed by the existing metabolite databases during training, their capacity to compensate for database incompleteness is not unlimited as observed with our LMSD-informed models having limited efficacy at higher mass ranges. The inclusion of additional sources of empirically observed lipids in these mass ranges may extend the useful mass range of this methodology. LMISSD-informed models did not suffer from this limitation but had decreased accuracy and specificity, potentially attributable to unrealistic entries and/or the highly unbalanced nature of the training set.

The observation that these models perform robustly on both training data and molecular formulas derived from spectra of real-world samples demonstrates the potential for real-world applications of these models. As demonstrated previously, one potential application of these models is the removal of molecular formulas after assignment that are unlikely to correspond to lipids before downstream analysis. Additionally, the predicted lipid categories and classes provided by these models enable differential abundance analyses to be performed on direct infusion MS1 data without the need for orthogonal information or tandem MS aiding assignment. Furthermore, since differential abundance analysis identifies differences at the lipid profile level rather than at the individual assignment level, the effect of occasional misassignment or incorrect lipid category prediction is minimized. Although our current models focus on lipid category prediction, similar models can foreseeably be created for any class of metabolite provided sufficient training data exists.

Thus, machine learning-based approaches will allow for more untargeted lipid profiling analyses than existing database-centric methods, even with the more limited data that can be acquired using direct injection MS1 alone. Similar methods could be applied to the classification of other major types of biomolecules or to identify potential contaminants or non-biological compounds detected in complex biological samples. However, the quality of the predictions made by such methods are fundamentally limited by the ability to generate high-quality molecular formula assignments in an unbiased manner for higher *m/z* ranges. Methods such as SMIRFE combined with cross-sample correspondence provide an avenue for generating high-quality assignments for these higher *m/z* ranges that can be further filtered by metabolite classification itself.

## Figures and Tables

**Figure 1 metabolites-10-00122-f001:**
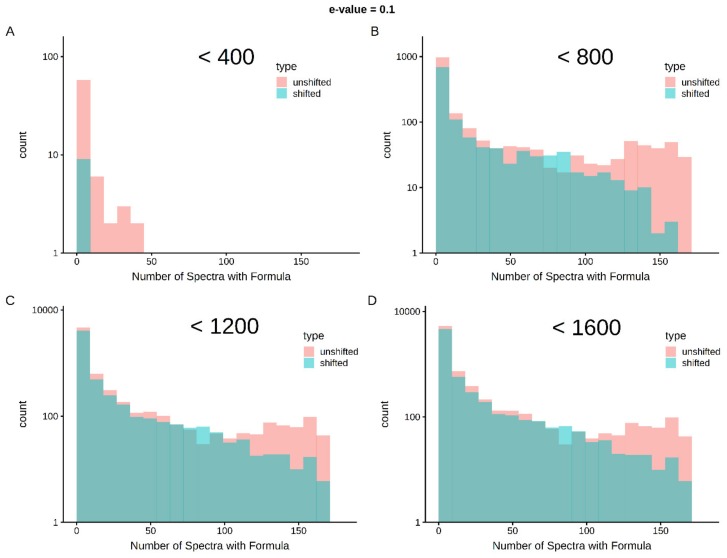
Cross-sample correspondence at E-value <= 0.1 identifies high-quality assignments with lipid classification—correct assignments are expected to occur more consistently within a set of samples than incorrect assignments. As shown in Panel **A**, below 400 *m/z*, very few assignments with lipid classification are made in the shifted spectra and very few of the assignments correspond across an appreciable number of spectra (i.e., the vast majority of the first bin represents single spectra assignments). As *m/z* increases (Panels **B**–**D**), shifted spectra have more assignments and by chance, some of these assignments correspond in multiple samples. However, at up to 1600 *m/z*, there are clearly more well-corresponding formulas in the unshifted assignments than in the shifted assignments.

**Figure 2 metabolites-10-00122-f002:**
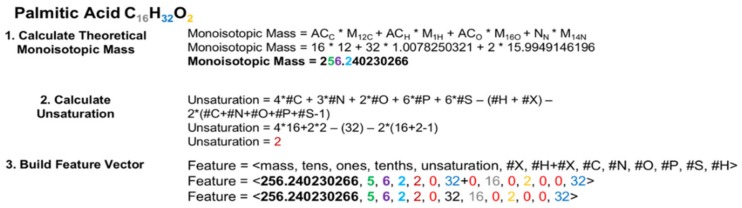
Example construction of a feature vector for the elemental molecular formula (EMF) C_16_H_32_O_2_, corresponding to palmitic acid. In a real-world application, the elemental molecular formula (EMF) would be provided from an assignment method, such as SMIRFE, and the compound it represents may not be known. The first step in constructing the feature vector is to calculate the theoretical monoisotopic mass for that EMF from the atom count for an element X (AC_x_) and the mass of its most naturally abundant isotope (M_x_). Calculating the theoretical mass for an EMF rather than relying upon the observed mass for the corresponding spectral feature eliminates the potential confound of mass error at the classification step. Calculating and using the monoisotopic mass is necessary so that isotopologues of the same EMF can be classified using the same classifiers. In the second step, the number of hydrogens missing in the formula due to unsaturation is calculated. Finally, the monoisotopic mass, the number of missing hydrogens, and the EMF are used to construct the feature vector. The coloring and bolding of the numbers in the example feature vector reflect the sources of these values.

**Figure 3 metabolites-10-00122-f003:**
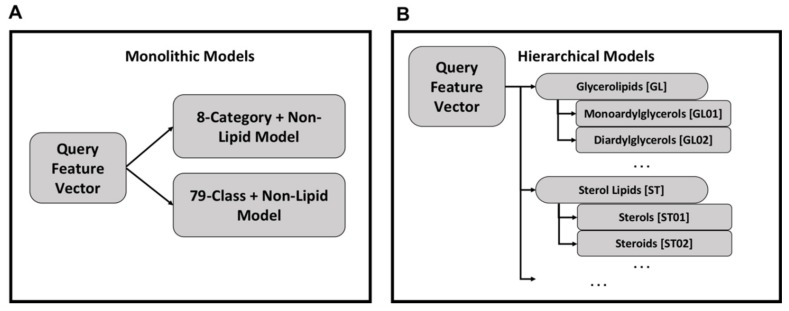
Organization of monolithic and hierarchical models—in a monolithic organization there exists one model for classifying feature vectors into lipid categories and another model for lipid classes (Panel **A**). This organization is simpler with fewer models to train compared to the hierarchical organization of models (Panel **B**). In the hierarchical organization, there is one model per category and class, with the class models organized under their respective category model.

**Table 1 metabolites-10-00122-t001:** LIPIDMAPS Structure Database (LMSD) + human metabolome database (HMDB)_non_lipid model performance (category).

LMSD + HMDB_non_Lipid Model Performance (Category)					
Category	Precision	Out-of-Bag Accuracy	Number of Entries	True Positives	False Positives
Fatty Acyls [FA]	0.838	0.901	2031	1681	324
Glycerolipids [GL]	0.996	0.995	532	520	2
Glycerophospholipids [GP]	0.995	0.996	1886	1886	10
Polyketides [PK]	0.780	0.884	1376	954	269
Prenol Lipids [PR]	0.989	0.970	473	263	3
Saccharolipids [SL]	1.0	0.998	102	99	0
Sphingolipids [SP]	0.996	0.993	1404	1386	6
Sterol Lipids [ST]	0.935	0.972	824	707	49
not_lipid	0.930	0.798	7587	6830	513

**Table 2 metabolites-10-00122-t002:** LMSD + LIPIDMAPS In-Silico Structure Database (LMISSD) + HMDB_non_lipid model performance (category).

LMSD + LMISSD + HMDB_non_Lipid Model Performance (Category)					
Category	Precision	Out-of-Bag Accuracy	Number of Entries	True Positives	False Positives
Fatty Acyls [FA]	0.837	0.939	2031	1659	322
Glycerolipids [GL]	0.995	0.993	2715	2696	14
Glycerophospholipids [GP]	0.979	0.979	9766	9706	206
Polyketides [PK]	0.768	0.933	1376	979	295
Prenol Lipids [PR]	0.985	0.983	473	259	4
Saccharolipids [SL]	1.000	0.998	102	99	0
Sphingolipids [SP]	0.976	0.976	3089	2875	72
Sterol Lipids [ST]	0.935	0.983	824	702	49
not_lipid	0.928	0.882	7587	6845	532

**Table 3 metabolites-10-00122-t003:** LMSD + HMDB_non_lipid model performance for convex hull (category).

LMSD + HMDB_non_lipid Model Performance for Convex Hull (Category)		
Category	Predictions	% of Hull Formulas
Fatty Acyls [FA]	475,516	0.429
Glycerolipids [GL]	8205	0.007
Glycerophospholipids [GP]	1,145,418	1.033
Polyketides [PK]	84,333	0.076
Prenol Lipids [PR]	18,684	0.016
Saccharolipids [SL]	6708	0.006
Sphingolipids [SP]	7,494,579	6.761
Sterol Lipids [ST]	18,643	0.017
not_lipid	74,621,680	67.31
no category	29,202,459	26.34

**Table 4 metabolites-10-00122-t004:** LMSD + LMISSD +HMDB_non_lipid model performance for convex hull (category).

LMSD + LMISSD +HMDB_non_lipid Model Performance for Convex Hull (Category)		
Category	Predictions	% of Hull Formulas
Fatty Acyls [FA]	393,314	0.354
Glycerolipids [GL]	56,116	0.051
Glycerophospholipids [GP]	1,735,925	1.566
Polyketides [PK]	118,968	0.107
Prenol Lipids [PR]	15,881	0.014
Saccharolipids [SL]	2795	0.002
Sphingolipids [SP]	12,568,226	11.34
Sterol Lipids [ST]	15,670	0.014
not_lipid	73,562,707	66.36
no category	27,808,607	25.08

**Table 5 metabolites-10-00122-t005:** LMSD + HMDB_non_lipid model performance for unshifted assignments.

LMSD + HMDB_non_lipid Model Performance for Unshifted Assignments		
Category	Predictions	% of Assigned Formulas
Fatty Acyls [FA]	639	0.502
Glycerolipids [GL]	795	0.624
Glycerophospholipids [GP]	8062	6.331
Polyketides [PK]	28	0.022
Prenol Lipids [PR]	1054	0.827
Saccharolipids [SL]	166	0.130
Sphingolipids [SP]	21,586	16.952
Sterol Lipids [ST]	358	0.281
not_lipid	54,389	42.71
no category	40,683	31.95

**Table 6 metabolites-10-00122-t006:** LMSD + HMDB_non_lipid model performance for shifted assignments.

LMSD + HMDB_non_lipid Model Performance for Shifted Assignments		
Category	Predictions	% of Assigned Formulas
Fatty Acyls [FA]	258	0.1951
Glycerolipids [GL]	923	0.7001
Glycerophospholipids [GP]	9517	7.227
Polyketides [PK]	37	0.0281
Prenol Lipids [PR]	1160	0.8808
Saccharolipids [SL]	233	0.1769
Sphingolipids [SP]	22,370	16.99
Sterol Lipids [ST]	257	0.1952
not_lipid	51,863	39.38
no category	45,663	34.67

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
