# Peer review of "Deriving Lipid Classification Based on Molecular Formulas"

_metabolites, 2020, doi:10.3390/metabo10030122_

Round 1
Reviewer 1 Report
Although the topic is not ground breaking news in lipidomics, the method described for assigning lipid classes by MS1 data is sound and valid. It could be of potential interest to researchers who need fast and uncomplicated analysis by MS1 for just getting a rough overview of the lipid classes contained in a biological sample. As the authors correctly state, any deeper assignment of molecular species will almost certianly require fragment spectra, and therefore the usability of this method might be rather limited. The manuscript itself is well written and comprehensive with nothing to add, the experimental data are sound and the conclusions are adaequate.
Author Response
Reviewer 1:
Although the topic is not ground breaking news in lipidomics, the method described for assigning lipid classes by MS1 data is sound and valid. It could be of potential interest to researchers who need fast and uncomplicated analysis by MS1 for just getting a rough overview of the lipid classes contained in a biological sample. As the authors correctly state, any deeper assignment of molecular species will almost certianly require fragment spectra, and therefore the usability of this method might be rather limited. The manuscript itself is well written and comprehensive with nothing to add, the experimental data are sound and the conclusions are adaequate.
Response:
We thank the reviewer for recognizing our effort to provide thorough description and evaluation of these methods that are appropriate for MS1 datasets. Even if the results are not “ground-breaking”, we have tried to perform high-quality research.
Reviewer 2 Report
The author of the study "Deriving Lipid Classification based on Molecular Formula" describe the application of the SMIRFE algorithm to retrieve lipid Ids via accurate mass determination / elemental composition and machine learning linked to the lipidmap database.
As an experimental scientist with experience in developing software and informatics concepts for lipidomics and mass spectrometry, I have problems to see the benefit for the proposed approach.
I do not feel well enough informed from the authors to judge tha soundness informatics behind the study. That might be a problem for a journal like metabolites and in the style how the informatics is described. And I would encourage to improve description of methods and concepts.
Basic tests often performd in OMICS were not presented. For determinations of false discovery rates from samples / data sets not containing any lipid. Or the deterministic test if from a real life experiments validated IDs (Golden Standards) would be detected. Assignment of 127,338 unique formulas for the human cancer sample is from my perspective not a good result and in my experience rather far away from the realistic number of lipids that can be detected in such experiments. Furthermore, the large number of isomers for certain lipid classes with the same elemental composition is not even adressed proper here. What is now the benefit?
In the introduction, basic work on utilization of high resolution MS in lipidomics is completly ignored as well as the concept of lipdomics screens described approximately ten years ago.
Author Response
Reviewer 2:
The author of the study "Deriving Lipid Classification based on Molecular Formula" describe the application of the SMIRFE algorithm to retrieve lipid Ids via accurate mass determination / elemental composition and machine learning linked to the lipidmap database.
As an experimental scientist with experience in developing software and informatics concepts for lipidomics and mass spectrometry, I have problems to see the benefit for the proposed approach.
I do not feel well enough informed from the authors to judge tha soundness informatics behind the study. That might be a problem for a journal like metabolites and in the style how the informatics is described. And I would encourage to improve description of methods and concepts.
Response:
First, we want to thank the reviewer for slogging their way through this review, even though they do not feel fully qualified to evaluate the informatics methods utilized. Any issue you point out due to a lack of clarity in manuscript becomes an important issue for us to address in order to improve the accessibility of the manuscript.
Also, we understand the reviewer’s perspective. The range of mass spectrometry methods for studying lipidomics has broadened. A lot of the field is focused on utilizing LC-MS and LC-MS/MS; however, the lipid assignment methods for these approaches are limited to matching to standards and/or databases of known/theoretic lipid compounds. We are trying to show what can be done in a highly untargeted approach, utilizing MS1 spectra from high-end Fourier transform mass spectrometry only. This overall approach can provide a quick overview of the lipid categories and classes that are present in a biological sample. We have tried to highlight these points in the introduction with the following changes:
(page 3, line 133) “In this manuscript, we present a novel method of predicting lipid category and class from molecular formula and information readily available from direct-infusion MS1 spectra without a reliance on existing metabolite databases or orthogonal information.”
Issue 1:
Basic tests often performd in OMICS were not presented. For determinations of false discovery rates from samples / data sets not containing any lipid. Or the deterministic test if from a real life experiments validated IDs (Golden Standards) would be detected. Assignment of 127,338 unique formulas for the human cancer sample is from my perspective not a good result and in my experience rather far away from the realistic number of lipids that can be detected in such experiments. Furthermore, the large number of isomers for certain lipid classes with the same elemental composition is not even adressed proper here. What is now the benefit?
Response:
We have already validated the overall SMIRFE method using a gold standard dataset in our 2019 Analytical Chemistry paper (Mitchell et al., 2019). The problem we run into with real non-polar biological samples is having a gold standard for assignment validation. Also, SMIRFE is a comprehensive search of molecular formula space. The 127,338 isotopically-resolved molecular formula assignments derived from 80 matched pair tumor and non-tumor human samples represents all possible assignments given the ambiguity in the data itself and no threshold for assignment quality. This was the starting point for evaluating if lipid classification can reduce assignment ambiguity when combined with assignment quality thresholds (i.e. E-value thresholds). We have added the following statements to make this point clearer:
(page 14, line 619) “Assignments were generated for 192 samples up to 1600 m/z and SMIRFE assigned a total of 127,338 unique formulas. This is a large number of assignments; however, this includes multiple possible assignments for each peak and does not account for any filtering of assignments based on their E-values.”
We have also added a description of the isomer content of the LIPIDMAPS and HMDB databases utilized in the classifier training.
(page 3, line 159) “As shown previously [38], isomerism is common with all metabolites but especially with lipids. This is reflected in both the LMSD and LMISSD which contain only 7.4% and .053% non-isomeric entries. See Supplemental Tables S1 and S2 for the level of isomerism broken down by category and class in LMSD and LMISSD, respectively. In the case of the LMISSD, this high amount of isomerism likely reflects both the high isomerism of lipids as a metabolite class as well as the methods used to generate the additional entries for the database in silico. Moreover, de-duplicating isomeric formulas is a necessary step in constructing a training dataset in order to prevent deleterious training effects from duplicate molecular formula entries.”
This will give some insight into the likely isomer content of biologically-based elemental molecular formula. Some of this information was provided previously in supplemental but we have made it more clear in the main text. We have also added a reference to a prior publication (Mitchell et al, 2013) that evaluated isomer content of KEGG and HMDB.
Issue 2:
In the introduction, basic work on utilization of high resolution MS in lipidomics is completly ignored as well as the concept of lipdomics screens described approximately ten years ago.
Response:
We have expanded the introduction to better represent the utilization of high resolution MS in lipidomics. However, this is not a review of the field, but an introduction to a FT-MS1 use-case that can provide quick information about the lipid categories and classes present in a biological sample:
(page 2, line 69) “Notably, improvements in FT-MS have enabled significant advances in shotgun lipidomics by distinguishing and quantifying isobaric lipids in place of in-depth MS/MS [30]. When combined with MS/MS, the analytical advantages of FT-MS enable global lipidome analysis and simultaneous structural characterization of some detected lipids [31]. However, the full utilization of the analytical capabilities of FT-MS for lipidomics, especially untargeted lipidomics, will require the development of new data analysis methods better tailored to the data provided by FT-MS.”
Reviewer 3 Report
This manuscript describes the details of combining a previously published algorithm with machine learning models to assign lipid classifications (8 lipid categories and 79 distinct classes) using MS data only. The authors describe an experiment in which the developed method (use of SMIRFE algorithm plus random forest machine learning model) was performed on a lung cancer data set. After appropriately training the models using known lipid and non-lipid specific datasets (lipid maps, in-silico lipid maps and HMDB after removing the lipid species), the data was used to determine the categories individual m/z’s were assigned to by using unshifted and shifted by 21 m/z. It seems that this manuscript is the next logical manuscript after release of the SMIRFE algorithm in Analytical Chemistry last year.
The idea of using MS data only to generate useful biological information about the samples with as little information as possible is intriguing and is important work for the lipidomic community and would be of interest to readers of Metabolites, therefore it is recommended that this work be published, after updated the document to reflect changes below.
- (page 1, line 16 and page 2, line 87) SMIRFE is not defined in this particular manuscript. A brief sentence overview and explanation of SMIRFE would be helpful for all readers (prior to getting to page 13, where there is a more in-depth discussion). Please update citation #25 to a complete citation.
- (page 8, line 278, etc.) The mass range chosen for these studies was m/z 150-1600, please address the spectral accuracy of the orbitrap mass analyser (in regards to the isotope ratios) for this mass range (as it is extremely important for this work). Recent study in analytical chemistry by Muddiman et. al. may need to be cited.
Khodjaniyazova S, Nazari M, Garrard KP, Matos MPV, Jackson GP, Muddiman DC. Characterization of the Spectral Accuracy of an Orbitrap Mass Analyzer Using Isotope Ratio Mass Spectrometry. Analytical Chemistry. 2018;90(3):1897-906. doi: 10.1021/acs.analchem.7b03983.
- The authors should add a paragraph to discuss ion suppression, as it is particularly important in the workflow (direct infusion analysis) that is used to generate the data in this manuscript.
(page 16, line 655) The appendix section A.2. outlines that MS/MS data was acquired for these runs. Is it possible to use this data to determine the validity of the machine learning hierarchical models? This would be extremely helpful to validate the data generated from the models. Another options would be to generate a ‘complex’ mixture of lipids from standards (the reviewer does realize that this option is complicated), analyze the samples and apply the SMIRFE algorithm and models to address their validity.
- (page 10, Mass Error and Classification Results and page 6, Multi-Classifier Performance on Experimentally-Observed Molecular Formulas) Was any new information/ biological inference learned from analyzing the NSCLC data set? If yes, that realization is unclear. It seems that the only information gleened from this experiment is that having an offset of 21m/z for each mass still results in numerous formulas to still be determined and thus lipid categories and classes to be ‘identified’? There is a sentence on page 6, last paragraph (lines 233-236) – “This result implies that the lipid classifier cannot be used alone to screen out all bad assignments when lipids are expected, instead other orthogonal data must be used to verify the quality of the assignments and select the correct assignments.” Please expand this discussion (in the appropriate sections) and particularly outline how the use of the developed model can be useful for biological interpretation if/when implemented. Until then, it is not clear that MS1 data can be used without orthogonal information.
- (pages 4-7) Add comma’s to all tables at the thousands, etc. Or at least be consistent in the use of comma’s. for Tables 1A-3B.
- (page 7, line 242) – Recommend defining NSCLC as it is not defined yet.
Author Response
Reviewer 3:
This manuscript describes the details of combining a previously published algorithm with machine learning models to assign lipid classifications (8 lipid categories and 79 distinct classes) using MS data only. The authors describe an experiment in which the developed method (use of SMIRFE algorithm plus random forest machine learning model) was performed on a lung cancer data set. After appropriately training the models using known lipid and non-lipid specific datasets (lipid maps, in-silico lipid maps and HMDB after removing the lipid species), the data was used to determine the categories individual m/z’s were assigned to by using unshifted and shifted by 21 m/z. It seems that this manuscript is the next logical manuscript after release of the SMIRFE algorithm in Analytical Chemistry last year.
The idea of using MS data only to generate useful biological information about the samples with as little information as possible is intriguing and is important work for the lipidomic community and would be of interest to readers of Metabolites, therefore it is recommended that this work be published, after updated the document to reflect changes below.
Response:
We thank the reviewer for supporting our practical, but logical approach of development and publication. We do feel that developing an approach to derive new biochemical knowledge based on molecular formula is the next logical step after SMIRFE. Therefore, we have focused this manuscript on describing and evaluating a machine learning approach for deriving lipid category and class from the molecular formulas generated from SMIRFE. We have striven to address each of the issues we have raised.
Issue 1:
(page 1, line 16 and page 2, line 87) SMIRFE is not defined in this particular manuscript. A brief sentence overview and explanation of SMIRFE would be helpful for all readers (prior to getting to page 13, where there is a more in-depth discussion). Please update citation #25 to a complete citation.
Response:
We have an overview and explanation of SMIRFE as follows:
(page 2, line 100) “Our methodology searches a near-exhaustive CHONPS elemental formula search space to find possible molecular formulas represented in a spectrum and then filters to a set of likely assignments by comparing observed intensity ratios between isotopologues of those molecular formulas [25].”
Issue 2:
(page 8, line 278, etc.) The mass range chosen for these studies was m/z 150-1600, please address the spectral accuracy of the orbitrap mass analyser (in regards to the isotope ratios) for this mass range (as it is extremely important for this work). Recent study in analytical chemistry by Muddiman et. al. may need to be cited.
Khodjaniyazova S, Nazari M, Garrard KP, Matos MPV, Jackson GP, Muddiman DC. Characterization of the Spectral Accuracy of an Orbitrap Mass Analyzer Using Isotope Ratio Mass Spectrometry. Analytical Chemistry. 2018;90(3):1897-906. doi: 10.1021/acs.analchem.7b03983.
Response:
We use a different approach that directly calculates the variance in the log isotopologue intensity ratios based on the observed ratio across multiple scans. Therefore, we do not talk about the accuracy of this ratio, rather the observed variance in the observed log ratio across many scans. Also, we derive this variance for each log intensity ratio. We have added these details into the dataset description (see response to issue 3), but also cite the paper to highlight the spectrum-level evaluation versus our scan-level evaluation.
Issue 3:
The authors should add a paragraph to discuss ion suppression, as it is particularly important in the workflow (direct infusion analysis) that is used to generate the data in this manuscript.
Response:
We discussed ion suppression in the SMIRFE Analytical Chemistry paper. We have added a statement with a reference to the SMIRFE paper:
(page 13, line 605) “To account for sources of intensity error, such as ion suppression and scan-to-scan injection efficiency differences, when SMIRFE compares the intensity of two peaks in a spectrum, it calculates a median intensity ratio across all the scans containing both peaks. Additionally, the log variance on that ratio across scans is calculated and used to inform the comparison of the observed intensity ratio to predicted [25].”
Issue 4:
(page 16, line 655) The appendix section A.2. outlines that MS/MS data was acquired for these runs. Is it possible to use this data to determine the validity of the machine learning hierarchical models? This would be extremely helpful to validate the data generated from the models. Another options would be to generate a ‘complex’ mixture of lipids from standards (the reviewer does realize that this option is complicated), analyze the samples and apply the SMIRFE algorithm and models to address their validity.
Response:
We have already validated SMIRFE in the Analytical Chemistry paper with a complex mixture of amino acids that have undergone chemoselection. Our focus in this manuscript is to describe our machine learning method and evaluate the lipid classification performance of the hierarchical machine classifiers.
Issue 5:
(page 10, Mass Error and Classification Results and page 6, Multi-Classifier Performance on Experimentally-Observed Molecular Formulas) Was any new information/ biological inference learned from analyzing the NSCLC data set? If yes, that realization is unclear. It seems that the only information gleened from this experiment is that having an offset of 21m/z for each mass still results in numerous formulas to still be determined and thus lipid categories and classes to be ‘identified’? There is a sentence on page 6, last paragraph (lines 233-236) – “This result implies that the lipid classifier cannot be used alone to screen out all bad assignments when lipids are expected, instead other orthogonal data must be used to verify the quality of the assignments and select the correct assignments.” Please expand this discussion (in the appropriate sections) and particularly outline how the use of the developed model can be useful for biological interpretation if/when implemented. Until then, it is not clear that MS1 data can be used without orthogonal information.
Response:
Again, our focus is on the lipid classification, not the performance of SMIRFE. However, we have elaborated on the discussion about how the lipid classification will be used in a downstream analysis that robustly provides lipid category and class assignments:
(page 15, line 699) “The observation that these models perform robustly on both training data and on molecular formulas derived from spectra of real-world samples demonstrates the potential for real-world applications of these models. As demonstrated previously, one potential application of these models is the removal of molecular formulas after assignment that are unlikely to correspond to lipids before downstream analysis. Additionally, the predicted lipid categories and classes provided by these models enable differential abundance analyses to be performed on direct infusion MS1 data without the need for orthogonal information or tandem-MS aiding assignment. Furthermore, since differential abundance analysis identifies differences at the lipid profile level rather than at the individual assignment level, the effect of occasional misassignment or incorrect lipid category prediction is minimized. Although our current models focus on lipid category prediction, similar models can foreseeably be created for any class of metabolite provided sufficient training data exists.” (
Please be patient with us. Deriving new biochemical knowledge in a highly untargeted manner requires a complex data analysis pipeline and a lot of methods development. We are trying to describe and fully validate each part of a complex data analysis pipeline in small enough chunks that readers can digest and understand. Many of the methods have other use-cases than just the one we are directly developing for.
Issue 6:
(pages 4-7) Add comma’s to all tables at the thousands, etc. Or at least be consistent in the use of comma’s. for Tables 1A-3B.
Response:
Fixed.
Issue 7:
(page 7, line 242) – Recommend defining NSCLC as it is not defined yet.
Response:
Fixed.
Round 2
Reviewer 2 Report
The answers to the usage of high resolution MS in lipidomics shows that there is lack of knowledge on the current state in the field.
But most concerning is the fact that my direct question regarding applicability and usefullness of the tool was not adressed at all.
At this stage I would reject the manuscript.